nanotechnology

ultra-fine silicon, Si nanoparticle, trichlorosilane, synthesis

**Author for correspondence:**
Mark H. Rümmeli
e-mail: mhr1967@yahoo.com

This article has been edited by the Royal Society of Chemistry, including the commissioning, peer review process and editorial aspects up to the point of acceptance.

# Facile production of ultra-fine silicon nanoparticles

Klaudia Tokarska[1], Qitao Shi[2], Lukasz Otulakowski[1], Pawel Wrobel[1], Huy Quang Ta[3], Przemyslaw Kurtyka[4], Aleksandra Kordyka[1], Mariola Siwy[1], Margaryta Vasylieva[1], Aleksander Forys[1], Barbara Trzebicka[1], Alicja Bachmatiuk[1,2,3] and Mark H. Rümmeli[1,2,3,5]

[1]Centre of Polymer and Carbon Materials, Polish Academy of Sciences (CMPW PAN), M. Curie-Sklodowskiej 34, Zabrze 41-819, Poland
[2]Soochow Institute for Energy and Materials Innovations (SIEMIS), College of Energy, Key Laboratory of Advanced Carbon Materials and Wearable Energy Technologies of Jiangsu Province, Soochow University, Suzhou 215006, People's Republic of China
[3]The Leibniz Institute for Solid State and Materials Research Dresden (IFW Dresden), Institute for Complex Materials, Helmholtzstrasse 20, 01069 Dresden, Germany
[4]Department of Biomaterials and Medical Devices Engineering, Faculty of Biomedical Engineering, Silesian University of Technology, Roosevelta 40, Zabrze 41-800, Poland
[5]Institute of Environmental Technology, VSB-Technical University of Ostrava, 17. Listopadu 15, Ostrava 708 33, Czech Republic

KT, 0000-0001-8796-5478; QS, 0000-0002-6881-6060;
PW, 0000-0001-6437-2485; PK, 0000-0001-8692-0737;
AK, 0000-0002-7549-5656; MS, 0000-0002-4341-1315;
AF, 0000-0002-6994-868X; BT, 0000-0002-6131-4017;
AB, 0000-0002-6547-3349; MHR, 0000-0003-3736-6439

A facile procedure for the synthesis of ultra-fine silicon nanoparticles without the need for a Schlenk vacuum line is presented. The process consists of the production of a $(HSiO_{1.5})_n$ sol–gel precursor based on the polycondensation of low-cost trichlorosilane ($HSiCl_3$), followed by its annealing and etching. The obtained materials were thoroughly characterized after each preparation step by electron microscopy, Fourier transform and Raman spectroscopy, X-ray dispersion spectroscopy, diffraction methods and photoluminescence spectroscopy. The data confirm the formation of ultra-fine silicon nanoparticles with controllable average diameters between 1 and 5 nm depending on the etching time.

# 1. Introduction

For almost 30 years, silicon nanoparticles (Si NPs) have attracted the interest of scientists due to their unique optoelectronic

properties, as well as their stable and diverse chemistry [1,2]. The advantages of silicon nanoparticles also include high natural abundance and their lack of biological toxicity [3]. Currently, one of the more desirable Si NPs in the technology industry is ultra-fine silicon nanoparticles (with diameters of less than 10 nm). They are attractive in microelectronic devices, for the conversion of solar energy, in light-emitting diodes (LEDs), photopumped tunable lasers and in sensors [3–9]. Ultra-fine silicon nanoparticles are also promising candidates as anode active material for lithium-ion batteries [10,11].

There are a number of methods for obtaining silicon nanoparticles. They include: purely physical processes such as pulsed laser ablation, heating degradation and ball milling [4,12]. Physico-chemical methods such as chemical techniques and the widely used electrochemical etching strategy are also possible [4]. These approaches typically usually yield nominal particle size along with narrow size distribution, scalable production and well-controlled surface chemistry [4,12].

Chemical synthesis routes are among the earliest reported strategies [13]. Ultra-fine silicon nanoparticles can be produced by reducing silicon halide $SiCl_4$ through reducing agents such as sodium, Zintl salts (ASi; A = K, Na, Mg), sodium naphthalenide and lithium aluminium hydride ($LiAlH_4$) in various surfactant solvents [2,4,7,8,13–20]. It has also been suggested that the use of inverse micelles, high-pressure and high-temperature bomb reactors or sonication are necessary to obtain ultra-fine Si nanoparticles [7,11,13,21,22]. Liu *et al.* developed a synthesis process based on the reaction of metal silicide ($Mg_2Si$, $NaSi$) with $Br_2$ or $NH_4Br$, also in the presence of a surfactant [9,23,24]. Brus *et al.* and Biesuz *et al.* based the production of crystalline silicon nanoparticles on a pyrolysis process [5,25]. There are also approaches based on solutions involving the reduction of a precursor or starting material, usually involving annealing and/or etching. Examples of starting materials for such a synthesis route are silicon monoxide (SiO) powders or sol–gel polymeric silicon precursors (silicates, polysiloxanes and silsesquioxanes) [1,3,10,26–29]. Silsesquioxanes are commercially widely available and easily processed in solution. Structurally, they are well-defined molecules composed of silicon–oxygen skeletons with the empirical formula ($RSiO_{1.5}$), in which R represents a variety of functional groups (e.g. H, alkyl, silyl and aromatic compounds). These functional groups have been widely studied and their chemistry is well established. In particular, hydrogen silsesquioxane (HSQ, $[HSiO_{3/2}]_n$), which can be easily processed and is stable, has gained considerable attention because of its earlier application as a spin-on dielectric in the microelectronics industry. The HSQ polymer precursor can be produced by the hydrolysis and polycondensation of trialkoxy- or trichlorosilane [3,6,26,27,30,31].

In this work, we present the production of ultra-fine silicon nanoparticles using a novel and economical technique using low-cost trichlorosilane ($HSiCl_3$) as the feedstock and without the need for a Schlenk vacuum line, which simplifies the process and is safer. The application of trichlorosilane ($HSiCl_3$) as a precursor allows one to obtain ultra-fine nanoparticles of controlled size, terminated by hydrogen bonding. This process is a scalable. The obtained final product matches the quality of Si nanoparticles prepared in similar synthetic routes by Veinot *et al.* [6,27] and Jaumann *et al.* [10]. Veinot *et al.* [27] obtained free-standing hydrogen-terminated silicon nanocrystals, as evidenced by the characteristic stretching and scissoring frequencies at *ca* 2100 and 910 cm$^{-1}$, respectively, from IR spectroscopic investigations. While, Jaumann *et al.* [10] obtained crystalline silicon nanoparticles with a size of 2–5 nm. They also noticed the tendency for the nanoparticles to agglomerate. Both groups also showed a small presence of oxygen in the obtained final product, which was attributed to residues and limited surface oxidation after sample preparation [6,10,27].

# 2. Experimental procedure

## 2.1. Chemicals

Trichlorosilane ($HSiCl_3$, 99%) and hydrofluoric acid for analysis (HF, 48–51% in water solution) were purchased from Acros Organics. Trichlorosilane ($HSiCl_3$) was stored at a temperature below 0°C. Ethyl alcohol absolute ($C_2H_5OH$, 99.8%) and hydrochloric acid (HCl, 35–38%) were purchased from POCH. All stock solutions were used as received.

## 2.2. Procedure

Before starting the synthesis, a Pyrex flask with a magnetic stirrer was sealed with a septum and flushed with argon for 2 h. Fifteen minutes before the end the flask was placed in an ice bath (salt was added to ice to help maintain a lower temperature) and cooled to a temperature of −20°C. After that, 20 ml of

trichlorosilane (HSiCl$_3$) was slowly added to a three-neck round-bottom flask. The flask was purged with argon continuously. The system was equipped with a simple exhaust to neutralize the evolving HCl through an NaOH solution-filled gas-wash bottle. Then, 0.6 ml of pure ethanol was added. After 10 min, deionized (DI) water (15 ml) was very carefully injected while vigorously stirring into the cooled HSiCl$_3$ through a septum. After 1 h, the ice bath was removed and the flask was allowed to warm to room temperature under an argon flow and was maintained for 3 h to evaporate any side products away. The pre-dry final product—(HSiO$_{1.5}$)$_n$ precursor was transferred to an oven and dried at 80°C under vacuum for at least 12 h.

The white solid and dry precursor was then ground in agate mortar and then annealed at 1150°C in an argon atmosphere (flow: 1 l min$^{-1}$) for 2 h (heating rate: 50°C min$^{-1}$ to 800°C, then 12.5°C min$^{-1}$).

In the etching procedure, 1.5 g of silicon-silica composite obtain after annealing process was placed into a Teflon beaker with magnetic stirrer. In total, 15 ml of dionized (DI) water and 3 ml of HCl (35–38%) were added. After 5 min stirring, 7.5 ml of HF (48–51%) was added. The entire solution was vigorously stirred for 25 min in the dark. The solution was filtered and rinsed two times with water (2 × 10 ml) and once with ethanol (10 ml). The obtained material was dried overnight in vacuum oven and store under argon atmosphere. The etching procedure is fully scalable.

## 2.3. Characterization

The obtained products were studied using Fourier transform infrared spectroscopy (FTIR) measured with a Thermo Scientific Nicolet 6700 spectrometer with suppressed total reflection (ATR). A Smart Orbit with a diamond crystal was used for this purpose. Intensities of the spectra were normalized relative to the maximum recorded peak. Raman spectra were measured using a WITec Confocal Raman Microscope (532 nm, 4 mW). Scanning electron microscopy (SEM) was performed using FEI Quanta 250 at an accelerating voltage of 20 kV, equipped with an EDS detector (EDAX). Photoluminescence (PL) spectroscopy was measured using the Hitachi F-2500 spectrofluorometer. X-ray diffraction measurements were carried out with a Bruker D8 Advance diffractometer with Cu K$\alpha_1$ radiation, 1.5418 Å. The measurements were made in the range of 10–90° $2\theta$. In addition, transmission electron microscopy (TEM, FEI Titan$^3$) was used.

# 3. Results and discussion

The synthesis strategy of ultra-fine Si nanoparticles is based on the previously reported method by Veinot *et al*. [27] and the modified route by Jaumann *et al*. [10] in which the production of (HSiO$_{1.5}$)$_n$ sol-gel polymer used as the precursor is obtained by the polycondensation of trichlorosilane (HSiCl$_3$) and the addition of water and ethanol. The next step involves annealing of the (HSiO$_{1.5}$)$_n$ precursor at 1150°C to obtain a silica matrix. The ultra-fine silicon nanoparticles are then obtained by chemical etching of the resulting oxide matrix (figure 1$d$).

In this study, by avoiding the use of a Schlenk vacuum line during the first step of the process, we simplify synthesis apparatus, which not only reduces cost but also minimizes the risk of implosion or explosion without compromising the quality or quantity of the obtained ultra-fine Si nanoparticles. Moreover, the implementation of argon both during the precursor synthesis as well as in the annealing step is a significant safety advantage. The developed system is presented in figure 1$a$–$c$.

As a result of the SiNP production steps, we obtain two intermediate materials: a (HSiO$_{1.5}$)$_n$ polymer precursor and a silicon-silica composite, and ultimately ultra-fine silicon nanoparticles. The (HSiO$_{1.5}$)$_n$ polymer precursor in the form of a white powder as a result of thermal processing, is transformed into a dark brown solid. After the etching process, the material's colour changes to light brown (figure 2$a$). Structural changes are also confirmed by SEM imaging studies (figure 2$b$). The 'fluffy' structure of the (HSiO$_{1.5}$)$_n$ powder is sintered and yields a uniform silica matrix after annealing. HF acid etching modification affects the release of the nanocrystalline silicon particles and size. Higher magnification images (right-most panel of figure 2$a$) show the powder nature of the final SiNP product and under SEM at higher magnifications (right-most panel of figure 2$b$) a 'fluffy' product is seen and indicates agglomerated SiNPs.

The structure and composition of the obtained materials at each step of preparation were studied and compared using FTIR spectroscopy (figure 3$a$), and with Raman spectroscopy measurements. For the (HSiO$_{1.5}$)$_n$ precursor the FTIR spectrum shows a characteristic vibration at *ca* 2252 cm$^{-1}$, which corresponds to the Si-H stretching vibration [27]. Peaks were also recorded in the range 950–1250 cm$^{-1}$,

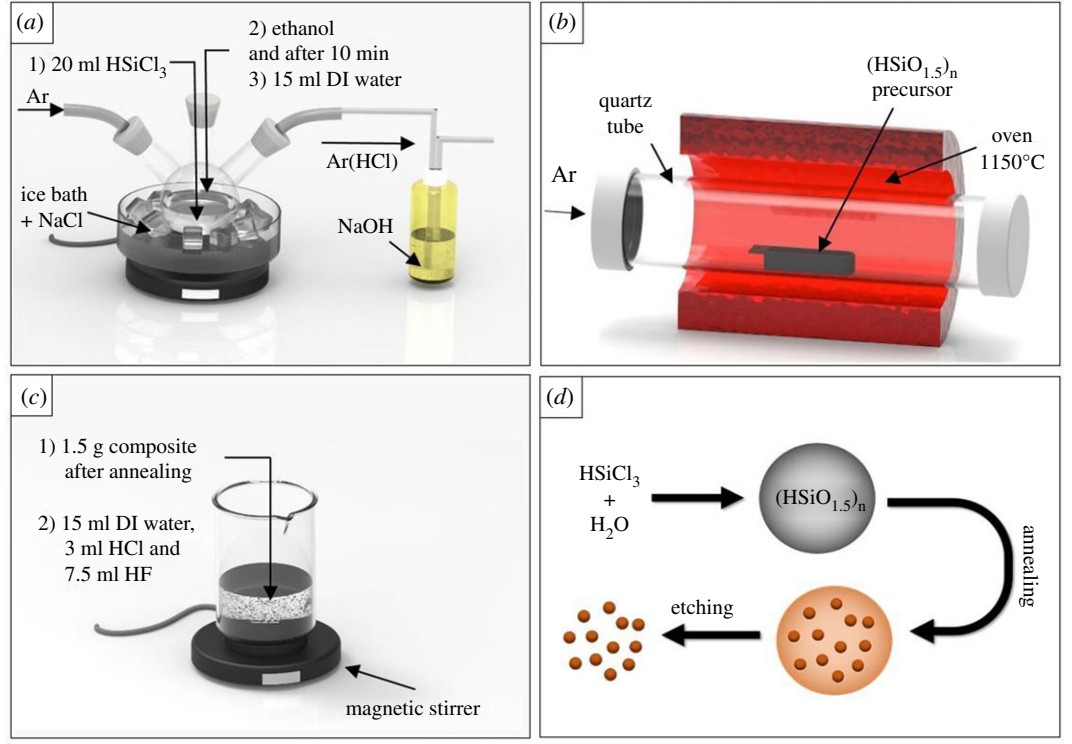

**Figure 1.** The SiNP preparation process consists of three steps: (*a*) $(HSiO_{1.5})_n$ condensation sol-gel polymer preparation; (*b*) annealing of $(HSiO_{1.5})_n$ precursor; (*c*) etching the oxide matrix to obtain free-standing silicon nanoparticles. (*d*) SiNP preparation strategy.

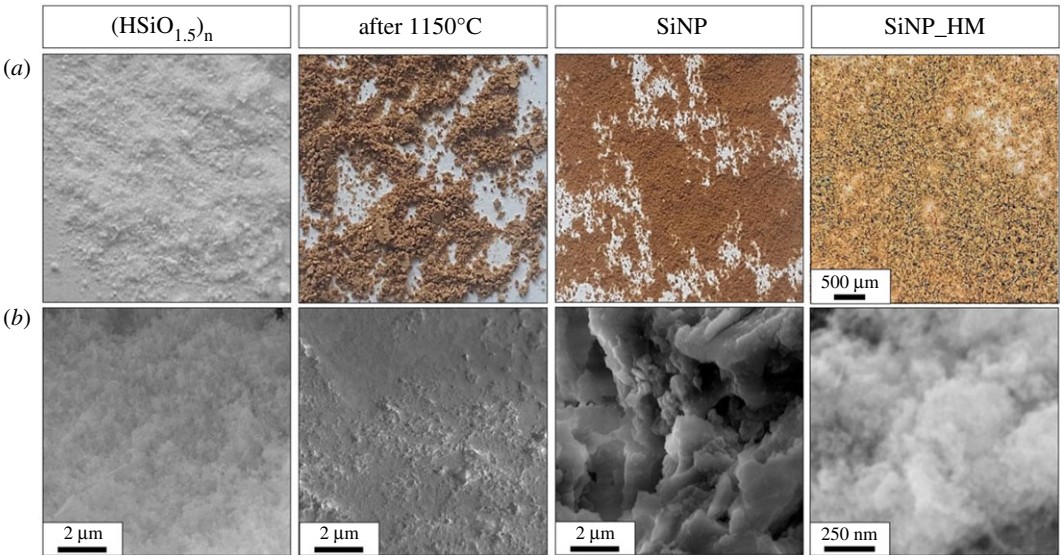

**Figure 2.** (*a*) Photos and (*b*) SEM images of materials after every production step. The right-most panels show higher optical magnifications (*a*) illustrating the powder nature of the SiNP product while the lower right-most panel (*b*) shows a fluffy material, indicating agglomerated SiNPs. Note: HM indicates higher magnification.

which we attribute to Si-O-Si bonds and recorded H-Si-O hybrid vibrations, whose centre is located at about $835 \, \text{cm}^{-1}$ [27,32]. After the precursor heating process at 1150°C, the following changes were observed: the Si-H stretching band (approx. $2252 \, \text{cm}^{-1}$) disappeared, the H-Si-O peak (approx. $835 \, \text{cm}^{-1}$) was significantly reduced, and the Si-O-Si vibration band (approx. $950-1250 \, \text{cm}^{-1}$) become sharper. These changes indicate the presence of a silicon-silica composite. For the ultra-fine Si nanoparticles obtained after etching (25 min), an Si-O-Si vibration band (approx. $950-1250 \, \text{cm}^{-1}$) is also observed, but its narrower peak width as compared with this peak for the silicon-silica composite suggests minor residual surface oxidation or incomplete etching [26,27]. The peak attributed to the H-Si-

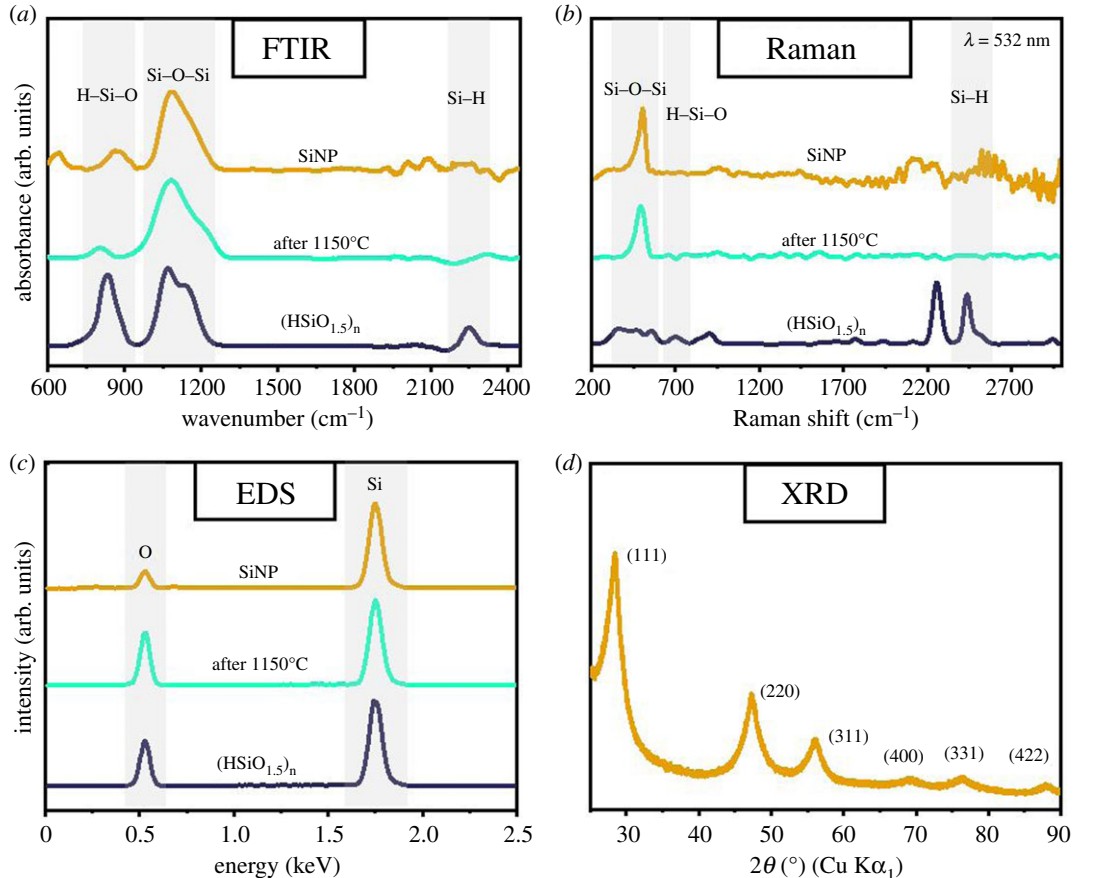

**Figure 3.** (*a*) FTIR spectrum, (*b*) Raman spectrum and (*c*) EDS analysis for SiNP, (HSiO$_{1.5}$)$_n$ after annealing at 1150°C and for (HSiO$_{1.5}$)$_n$ precursor; (*d*) XRD pattern of bare silicon nanoparticles after 25 min of etching.

O bond (approx. 835 cm$^{-1}$) increased and a weak noisy signal can be observed in the range of Si-H bond (approx. 2252 cm$^{-1}$), which may indicate partial hydrogen termination at silicon nanoparticle surface which would provide them with a degree of environmental stability [26,27]. Complementary data were found from the Raman spectroscopy (figure 3*b*). Peaks in the region of 500–550, 700 and 2440 cm$^{-1}$ were recorded for the (HSiO$_{1.5}$)$_n$ precursor, which can be attributed to Si-O-Si, H-Si-O and Si-H bonding configurations, respectively [33]. A sharp Si-O-Si peak at *ca* 505 cm$^{-1}$ is observed for the silicon nanoparticles [33,34]. EDX analysis also confirmed the presence of silicon (1.75 keV) and a weak oxygen peak (0.5 keV) (figure 3*c*). The presence of oxygen in the final product can be due in part to minute surface oxide formation and trapped oxygen species. This matches observations by others [6,10,27].

Figure 3*d* shows the X-ray powder diffraction pattern of silicon nanoparticles obtained after a 25 min etching period. The intense (2θ) diffraction peaks at approximately 28°, 47° and 56° can be easily attributed to the (111), (220) and (311) orientations, respectively, which are typical for crystalline cubic silicon [10,27,35]. Higher-order crystal orientations can also be observed [10,35–37]. These results agree with the published data and further confirm the formation of ultra-fine silicon nanocrystals [10,26,35,36].

Figure 4 shows low-magnification TEM micrographs of the ultra-fine silicon nanoparticles after 25 min, 85 min, 115 min and 135 min etching time in panels *a*, *d*, *e* and *f*, respectively. The TEM data show that the ultra-fine silicon nanoparticles tend to form agglomerates. This we attribute to the strong hydrophobic character of silicon hydride termination and their high surface energy [10]. Figures 4*c* and 5*c* show a high magnification micrographs of silicon nanoparticles, they show clear lattice fringes with a *d* spacing of approximately 0.3 and 0.24 nm corresponding to the (111) and (100) orientation of crystalline silicon [10,11,16,38]. The TEM studies also show that the SiNPs agglomerate easily; however, careful examination of the particles reveals that they are not faceted and are somewhat spherical in form as, for example, shown in false colour in figure 5*c,d*.

The mean size and size distribution of the as-produced ultra-fine Si nanoparticles was determined by measuring 500 nanoparticles from different regions on the grid. The average diameter of silicon

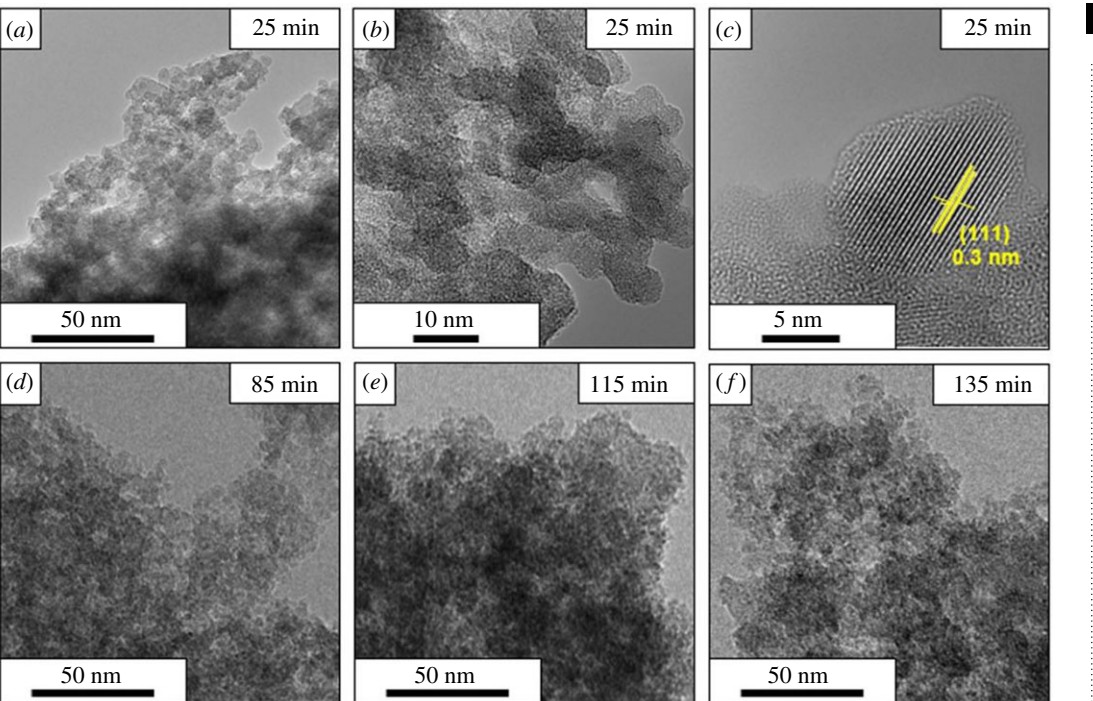

**Figure 4.** TEM images of SiNP after (*a*) 25 min, (*d*) 85 min, (*e*) 115 min and (*f*) 135 min of etching in low magnification; (*b*) medium and (*c*) high magnification of SiNP after 25 min of etching.

nanoparticles after 25 min of etching is 4.7 nm (±1.5 nm), which coincides with the Scherrer analysis from the X-ray diffractometry studies. The average size of the obtained crystallites after 25 min etching was estimated to be 4.8 nm (±0.9 nm) after the Scherrer analysis. Particle-size distribution histograms are shown in electronic supplementary material, figure S1. We also explored adjustment of the ultra-fine Si NPs mean size through etching time control. The size of nanoparticles decreases with longer etching times. Figure 6*a* shows the average size decrease for etching times of 25, 85, 115 and 135 min. The inset shows the full width at half maximum (FWHM) for the Si NPs size distributions for the different etching times. One sees a narrowing of the FWHM with increased etching time. The ultra-fine Si nanoparticles after different etching times still remain crystalline, as confirmed by XRD, FTIR and EDS measurements (electronic supplementary material, figure S2). Regarding the etching mechanisms, the processes involved have not yet been fully elucidated. To date, studies show that diluted HF digests the silicon, while concentrated HF (49%) leaves the crystalline silicon untouched [39]. The easiest approach to remove any native surface oxide is to rapidly immerse the material in diluted HF, which should not significantly change the surface morphology of the silicon [39]. HF acid attacks polarized Si-O bonds, thereby removing the oxide and leaving a passivated hydrogenated silicon surface. Raghavachari *et al*. in their research proposed a model in which some of the surface atoms are fluorinated, which allows for continuous digestion/etching of crystalline silicon [40–42]. Jacob *et al*. suggested that OH⁻ ions present in pH-enhanced etching solutions play an essential role in etching the Si (111) surface and probably limit the rate of silicon digestion [42,43].

PL measurements were carried out for the silicon nanoparticles after 25, 85, 115 and 135 min etching times. The PL spectra of Si nanoparticles suspended in toluene are shown in figure 6*b* [44]. The measurements were performed at room temperature using an excitation wavelength of 220 nm. For the tested samples, similar spectra with a peak in the 390–392 nm (± 1 nm) range were observed. This suggests that small changes in diameter from the ultra-fine Si NPs do not significantly alter their PL properties. The position of the PL peaks in the violet region of the spectrum is consistent with previous reports of silicon nanocrystals of similar size [7,8,19,22]. Wilcoxon *et al*. [22] for silicon nanoparticles in the size range 2–10 nm received the most intense PL with a peak centred at approximately 365 nm (excitation wavelength 245 nm), and Tilley *et al*. [7] for nanoparticles 1.8 nm ± 0.2 nm at an excitation wave of 290 nm observed a peak with a maximum at 335 nm. Both research groups attributed the obtained PL results to direct electron-hole recombination in the silicon nanocrystals, which can also be attributed to the lack of PL from defect or trap state recombination,

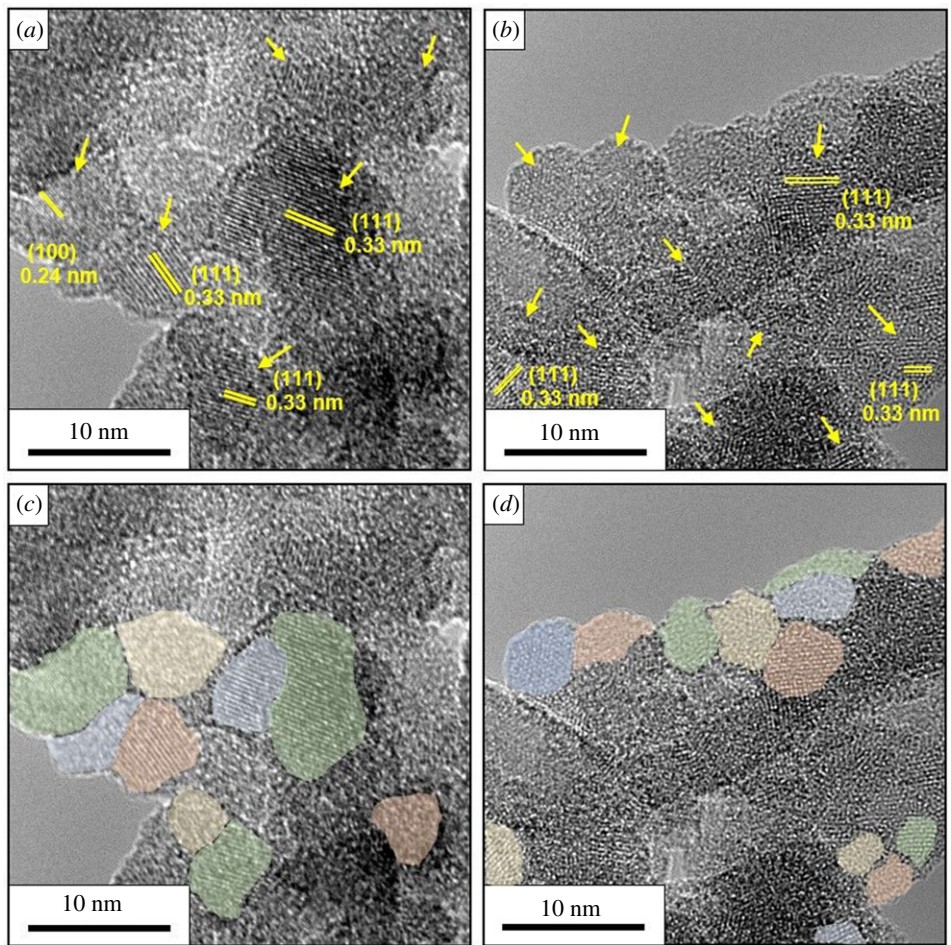

**Figure 5.** HR TEM images of nanocrystalline SiNP. Panels (*a,b*) show the agglomerated nature of the SINPs and that the SiNPs are crystalline. Lattice fringes and *d* spacings are indicated in yellow along with the orientations. Panels (*c,d*) highlight the shape of various SiNPs in false colour for panels *a* and *b*, respectively.

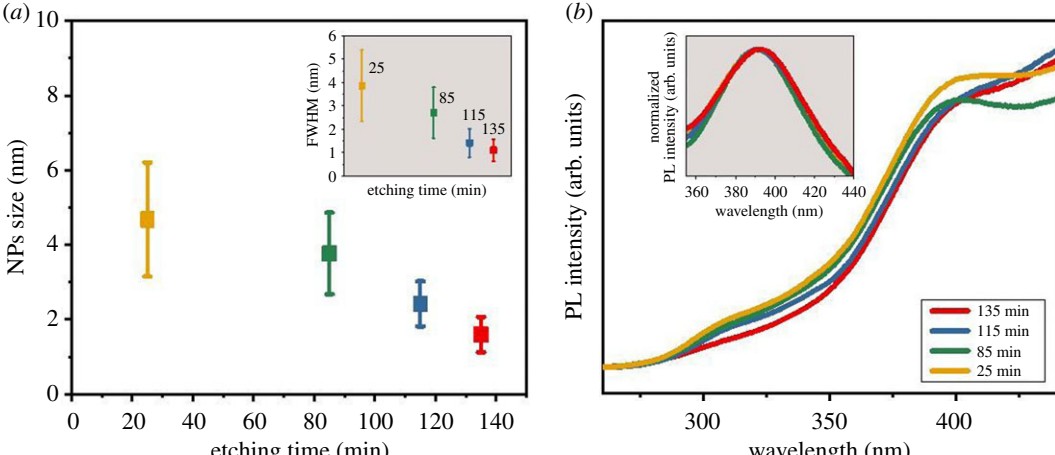

**Figure 6.** (*a*) Average silicon nanoparticle size after 25, 85, 115 and 135 min of etching (Inset: FWHM of silicon nanoparticles diameter profile); (*b*) PL spectra of toluene suspensions of SiNP after etching.

which usually occurs around 600 nm [7]. Different solvents, differences in the size range of the nanoparticles or their termination may also cause slight differences in the range of the PL peak [1,8,26,27,44].

We also noted that PL intensity depends on the concentration and time of sample preparation using ultrasonication. The following concentrations were tested: 0.25%, 0.1%, 0.05% and sonication times: 1, 5,

10, 20, 30 min. The strongest peak intensities were found for a 0.1% concentration and after 10 min of sonication, which is probably related to the disintegration of the silicon nanoparticle agglomerates [17,21,28].

# 4. Conclusion

In summary, we have successfully demonstrated a simplified synthesis protocol for ultra-fine silicon nanoparticles which has the advantage of being safer and cheaper than most other approaches. The technique could easily be scaled up. Spectroscopic and microscopy characterizations as well as diffractometric studies confirm the formation of crystalline ultra-fine SINPs with controllable mean diameters ranging from 1 to 5 nm. PL studies were conducted. The PL results point to direct electron-hole recombination in the silicon nanocrystals, which can also be attributed to the lack of PL from defect or trap state recombination. The presented synthesis approach for ultra-fine silicon nanoparticles helps advance their fabrication, which can be useful for applications, for example, SiNP as active anode material in lithium-ion batteries.

Data accessibility. Data supporting this paper are provided in the electronic supplementary material.
Authors' contributions. K.T. and Q.S. conducted the experiments; L.O. and P.W. contributed to the ideas for the first step of the synthesis process; A.K. made XRD measurements; M.S. and M.V. performed PL measurements; A.B., K.T. and A.F. carried out TEM measurements; K.T. performed SEM, EDX, FTIR and Raman measurements; H.Q.T. and P.K. helped in conducting researches and the data analysis; K.T. and M.H.R. wrote the paper with meritable support from B.T. All authors contributed to the general discussion and manuscript.
Competing interests. The authors declare that they have no competing interests.
Funding. This work was supported by the National Science foundation China (grant no. 51672181) and the Czech Republic from ERDF 'Institute of Environmental Technology – Excellent Research' (grant no. CZ.02.1.01/0.0/0.0/16_019/0000853).
Acknowledgements. M.H.R. thanks the Sino-German Research Institute for support (project: GZ 1400).

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
