## [Reviewer comments · Royal Society Open Science]

Review History

RSOS-200736.R0 (Original submission)

Review form: Reviewer 1

Is the manuscript scientifically sound in its present form?

Yes

Are the interpretations and conclusions justified by the results?

Yes

Is the language acceptable?

Yes

Do you have any ethical concerns with this paper?

No

Have you any concerns about statistical analyses in this paper?

No

Recommendation?

Accept with minor revision (please list in comments)

Comments to the Author(s)

The paper presents in a clear manner the efficient approach to the synthesis of silicon nanoparticles. The authors characterized the materials at every stage of the preparation and provided reasonable discussion of the results. The authors should take into account the following comment: the reason behind a decrease of the Si NPs size with increased etching time should be discussed.

Review form: Reviewer 2

Is the manuscript scientifically sound in its present form?

No

Are the interpretations and conclusions justified by the results?

No

Is the language acceptable?

Yes

Do you have any ethical concerns with this paper?

No

Have you any concerns about statistical analyses in this paper?

No

Recommendation?

Major revision is needed (please make suggestions in comments)

Comments to the Author(s)

The authors present a procedure of the synthesis of ultra-fine silicon nanoparticles. I have the following questions:

1. The demerits of this simplified process. The authors claim this route is simplify, safe and scalable. But how about the quality of the NPs? Specially, is it comparable with the quality of NPs prepared from similar routes from ref.10 and ref.27? This should be clarified in the introduction part.
2. For the chemicals used in the paper, I suggest the authors to use the chemical formulas instead of (or with) the chemical names used in the paper, which is easier to read. Besides, the concentration of the chemicals for synthesis should be given. For example, the concentrations of HF and HCl on page 3 lines 14-17 are missing.
3. It is critical that the SEM of SiNP (Fig 2) are NOT nanoparticles. Worse yet, the FTIR, Raman, and EDS all show Oxygen in the final products. The product is NOT pure Si. Even the high resolution TEM (Fig.4c) show one Si NP locally, and the XRD show Si Peaks, the quality of the products is really suspicious.

Based on these, the authors should optimize the process of synthesis and at least present Nanoparticles of Si.

Decision letter (RSOS-200736.R0)

Dear Dr Rummeli:

Title: Facile production of ultra-fine silicon nanoparticles
Manuscript ID: RSOS-200736

The editor assigned to your manuscript has now received comments from reviewers. We would like you to revise your paper in accordance with the referee and Subject Editor suggestions which can be found below (not including confidential reports to the Editor). Please note this decision does not guarantee eventual acceptance.

Please submit your revised paper before 18-Jul-2020. Please note that the revision deadline will expire at 00.00am on this date. If we do not hear from you within this time then it will be assumed that the paper has been withdrawn. In exceptional circumstances, extensions may be possible if agreed with the Editorial Office in advance. We do not allow multiple rounds of revision so we urge you to make every effort to fully address all of the comments at this stage. If deemed necessary by the Editors, your manuscript will be sent back to one or more of the original reviewers for assessment. If the original reviewers are not available we may invite new reviewers.

On behalf of the Subject Editor Professor Anthony Stace and the Associate Editor Dr Dattatray Late.

RSC Associate Editor:

Comments to the Author:

Detail morphological and structural characterization of silicon nanoparticles along with detail analysis need to be added. Interesting article to be considered in this journal.

RSC Subject Editor:

Comments to the Author:

Reviewers' Comments to Author:

Reviewer: 1

Comments to the Author(s)

The paper presents in a clear manner the efficient approach to the synthesis of silicon nanoparticles. The authors characterized the materials at every stage of the preparation and provided reasonable discussion of the results. The authors should take into account the following comment: the reason behind a decrease of the Si NPs size with increased etching time should be discussed.

Reviewer: 2

Comments to the Author(s)

The authors present a procedure of the synthesis of ultra-fine silicon nanoparticles. I have the following questions:

1. The demerits of this simplified process. The authors claim this route is simplify, safe and scalable. But how about the quality of the NPs? Specially, is it comparable with the quality of NPs prepared from similar routes from ref.10 and ref.27? This should be clarified in the introduction part.
2. For the chemicals used in the paper, I suggest the authors to use the chemical formulas instead of (or with) the chemical names used in the paper, which is easier to read. Besides, the concentration of the chemicals for synthesis should be given. For example, the concentrations of HF and HCl on page 3 lines 14-17 are missing.
3. It is critical that the SEM of SiNP (Fig 2) are NOT nanoparticles. Worse yet, the FTIR, Raman, and EDS all show Oxygen in the final products. The product is NOT pure Si. Even the high resolution TEM (Fig.4c) show one Si NP locally, and the XRD show Si Peaks, the quality of the products is really suspicious.

Based on these, the authors should optimize the process of synthesis and at least present Nanoparticles of Si.

Author's Response to Decision Letter for (RSOS-200736.R0)

See Appendix A.

Decision letter (RSOS-200736.R1)

Dear Dr Rummeli:

Title: Facile production of ultra-fine silicon nanoparticles
Manuscript ID: RSOS-200736.R1

It is a pleasure to accept your manuscript in its current form for publication in Royal Society Open Science. The chemistry content of Royal Society Open Science is published in collaboration with the Royal Society of Chemistry.

On behalf of the Subject Editor Professor Anthony Stace and the Associate Editor Dr Dattatray Late.

RSC Associate Editor
Comments to the Author:
Authors have revised the manuscript as per referee's suggestions and found them satisfactory.

Reviewer(s)' Comments to Author:

Appendix A

Response to Reviewers

We first wish to thank the Editor, their team and the Reviewers for their constructive efforts to improve our manuscript. We believe we were able to address all criticism and the new manuscript version is ready for publication. Below we address each of the Reviewer's comments (point by point). In addition, a revised manuscript with highlights indicating revisions is also provided.

I. RSC Associate Editor

Detail morphological and structural characterization of silicon nanoparticles along with detail analysis need to be added. Interesting article to be considered in this journal.

Response: We thank Editor for sharing this. We also believe this work can be interest to the scientific community. We agree some morphological detail and structural characteristics need to be improved. To this end, we have now conducted additional optical microscopy, SEM and TEM characterizations to better evaluate the material. In the revised manuscript, we have added a higher magnification optical micrograph confirming the powdery nature of the SiNP product (figure 2A right most panel) and we have also added a higher magnification SEM image which shows the “fluffy” like nature of the SiNPs which are in an agglomerated form (see figure 2B right most panel). We have also added new text to discuss the new data in figure 2. For convenience, the additional text and modified figure are provided below:

Additional text:

Higher magnification images (right most panel of 2A) show the powder nature of the final SiNP product and under SEM at higher magnifications (right most panel of 2B) a “fluffy” product is seen and indicates agglomerated SiNPs.

The modified Figure 2:

Figure 2. (A) Photos and (B) SEM images of materials after every production steps. The right most panels show higher optical magnifications (A) illustrating the powder nature of the SiNP product while the lower right most panel (B) shows a fluffy material, indicating agglomerated SiNPs. Note: HM indicates Higher Magnification

We have also prepared a new figure, Figure 5, in which additional TEM investigations reveal the SiNP nature, confirming they are crystalline and we also provide additional micrographs with false color highlighting the shape of the NPs. We have also added additional text in the manuscript. For ease they are provided below:

Additional text:

Panel C of figure 4 and figure 5 show a high magnification micrographs of silicon nanoparticles, they show clear lattice fringes with a d spacing ≈ 0.3 nm and $d \approx 0,24$ nm corresponding to the (111) and (100) orientation of crystalline silicon [10, 11, 16, 38]. The TEM studies also show that the SiNPs agglomerate easily, however careful examination of the particles reveals that they are not faceted and are somewhat spherical in form as, for example, shown in false colour in panels C and D of figure 5.

Additional Figure 5:

Figure 5. HR TEM images of nanocrystalline SiNP. Panels A and B show the agglomerated nature of the SiNPs and that the SiNPs are crystalline. Lattice fringes and d spacing's are indicated in yellow along with the orientations. Panels C and D shows the highlight the shape of various SiNPs in false color for panels A and B, respectively.

II. Reviewer #1

The paper presents in a clear manner the efficient approach to the synthesis of silicon nanoparticles. The authors characterized the materials at every stage of the preparation and provided reasonable discussion of the results. The authors should take into account the following comment: the reason behind a decrease of the Si NPs size with increased etching time should be discussed.

Response: We thank the reviewer for highlighting this point. In practice, the full details of the mechanisms involved are incomplete. However, some ideas are emerging and to this end we have now added the following into the manuscript to indicate recent research on the matter. It reads:

Regards the etching mechanisms, the processes involved have not yet been fully elucidated. To date studies, show that diluted HF digests the silicon, while concentrated HF (49%) leaves the crystalline

silicon untouched [39]. The easiest approach to remove any native surface oxide is to rapidly immerse the material in diluted HF, which should not significantly change the surface morphology of the silicon [39]. HF acid attacks polarized Si - O bonds, thereby removing the oxide and leaving a passivated hydrogenated silicon surface. Raghavachari et al. in their research proposed a model in which some of the surface atoms are fluorinated, which allows for continuous digestion/etching of crystalline silicon [40–42]. Jacob et al. suggested that OH⁻ ions present in pH-enhanced etching solutions play an essential role in etching the Si (111) surface and probably limit the rate of silicon digestion [42, 43].

III. Reviewer #2

The authors present a procedure of the synthesis of ultra-fine silicon nanoparticles. I have the following questions:

Question 1: *The demerits of this simplified process. The authors claim this route is simplify, safe and scalable. But how about the quality of the NPs? Specially, is it comparable with the quality of NPs prepared from similar routes from ref.10 and ref.27? This should be clarified in the introduction part.*

Response: We thank Reviewer for raising this point. To clarify this issue, we have expanded the introduction. For ease the relevant paragraph now reads as:

In this work, we present the production of ultra-fine silicon nanoparticles using a novel and economical technique using low-cost trichlorosilane (HSiCl₃) as the feedstock and without the need for a Schlenk vacuum line, which simplifies the process and is safer. The application of trichlorosilane (HSiCl₃) as a precursor allows one to obtain ultra-fine nanoparticles of controlled size, terminated by hydrogen bonding. This process is a scalable. The obtained final product matches the quality of Si nanoparticles prepared in similar synthetic routes by Veinot et al. [6, 27] and Jaumann et al. [10]. Veinot et al. [27] obtained free-standing hydrogen-terminated silicon nanocrystals, as evidenced by the characteristic stretching and scissoring frequencies at ca. 2100 and 910 cm⁻¹, respectively from IR spectroscopic investigations. While, Jaumann et al. [10] obtained crystalline silicon nanoparticles with a size of 2-5 nm. They also noticed the tendency for the nanoparticles to agglomerate. Both groups also showed a small presence of oxygen in the obtained final product, which was attributed to residues and limited surface oxidation after sample preparation [6, 10, 27].

Question 2: *For the chemicals used in the paper, I suggest the authors to use the chemical formulas instead of (or with) the chemical names used in the paper, which is easier to read. Besides, the concentration of the chemicals for synthesis should be given. For example, the concentrations of HF and HCl on page 3 lines 14-17 are missing.*

Response: We thank very much the Reviewer's comments and suggestions. We agree that the use of chemical formulas will be easier to read. We updated the chemical formulas in the manuscript. We did already mentioned the concentration of the chemicals for synthesis (HF and HCl) on page 2 lines 46-48. We also added the concentration of HF and HCl on page 3 lines 14-17. For the Reviewer's convenience the introduced changes are presented below:

2.1 Chemicals

Trichlorosilane (HSiCl_3 , 99%) and hydrofluoric acid for analysis (HF, 48 to 51% in water solution) were purchased from Acros Organics. Trichlorosilane (HSiCl_3) was stored at a temperature below 0°C . Ethyl alcohol absolute ($\text{C}_2\text{H}_5\text{OH}$, 99.8%) and hydrochloric acid (HCl, 35-38%) were purchased from POCH. All stock solutions were used as received.

2.2 Procedure

Before starting the synthesis, a Pyrex flask with a magnetic stirrer was sealed with a septum and flushed with argon for 2h. 15 min before the end the flask was placed in an ice bath (salt was added to ice to help maintain a lower temperature) and cooled to a temperature of -20°C . After that, 20 ml of trichlorosilane (HSiCl_3) was slowly added to a three-neck round-bottom flask. The flask was purged with argon continuously. The system was equipped with a simple exhaust to neutralize the evolving HCl through a NaOH solution filled gas-wash bottle. Then, 0.6 ml of pure ethanol was added. After 10 min, deionized (DI) water (15 ml) was very carefully injected while vigorously stirring into the cooled HSiCl_3 through a septum. After 1h the ice bath was removed and the flask was allowed to warm to room temperature under an argon flow and was maintained for 3h to evaporate any side products away. The pre-dry final product - $(\text{HSiO}_{1.5})_n$ precursor was transferred to an oven and dried at 80°C under vacuum for at least 12h.

The white solid and dry precursor was then ground in agate mortar and then annealed at 1150°C in an argon atmosphere (flow: 1L/min) for 2h (heating rate: $50^\circ\text{C}/\text{min}$ to 800°C , then $12.5^\circ\text{C}/\text{min}$).

In the etching procedure 1.5 g of silicon-silica composite obtain after annealing process was placed into a Teflon beaker with magnetic stirrer. 15 ml of dionized (DI) water and 3 ml of HCl (35-38%) were added. After 5 min stirring, 7.5 ml of HF (48-51%) was added. The entire solution was vigorously stirred for 25 min in the dark. The solution was filtered and rinsed two times with water (2 x 10 ml) and once with ethanol (10 ml). The obtained material was dried overnight in vacuum oven and store under argon atmosphere. The etching procedure is fully scalable.

Question 3: *It is critical that the SEM of SiNP (Fig 2) are NOT nanoparticles. Worse yet, the FTIR, Raman, and EDS all show Oxygen in the final products. The product is NOT pure Si. Even the high resolution TEM (Fig.4c) show one Si NP locally, and the XRD show Si Peaks, the quality of the products is really suspicious.*

Based on these, the authors should optimize the process of synthesis and at least present Nanoparticles of Si.

Response: We thank Reviewer for highlighting this points. We would like to break down this question into three parts:

- *It is critical that the SEM of SiNP (Fig 2) are NOT nanoparticles.*

We very much appreciate the Reviewer's comment. We apologize for any confusion. In the SEM photos we wanted to show how the material structure changes during the each stage of Si nanoparticles production at low magnification which gives an unclear message. In view of this we conducted additional optical and electron microscopy studies at higher magnifications.

In the revised manuscript, we have added a higher magnification optical micrograph confirming the powdery nature of the SiNP product (figure 2A right most panel) and we have also added a higher magnification SEM image which shows the "fluffy" like nature of the SiNPs which are in an agglomerated form (see figure 2B right most panel). We have also added new text to discuss the new data in figure 2. For convenience, the additional text and modified figure are provided below:

Additional text:

Higher magnification images (right most panel of 2A) show the powder nature of the final SiNP product and under SEM at higher magnifications (right most panel of 2B) a "fluffy" product is seen and indicates agglomerated SiNPs.

The modified Figure 2:

Figure 2. (A) Photos and (B) SEM images of materials after every production steps. The right most panels show higher optical magnifications (A) illustrating the powder nature of the SiNP product while the lower right most panel (B) shows a fluffy material, indicating agglomerated SiNPs. Note: HM indicates Higher Magnification

We have also prepared a new figure, Figure 5, in which additional TEM investigations reveal the SiNP nature, confirming they are crystalline and we also provide additional micrographs with false color highlighting the shape of the NPs. We have also added additional text in the manuscript. For ease they are provided below:

Additional text:

Panel C of figure 4 and figure 5 show a high magnification micrographs of silicon nanoparticles, they show clear lattice fringes with a d spacing ≈ 0.3 nm and $d \approx 0,24$ nm corresponding to the (111) and (100) orientation of crystalline silicon [10, 11, 16, 38]. The TEM studies also show that the SiNPs agglomerate easily, however careful examination of the particles reveals that they are not faceted and are somewhat spherical in form as, for example, shown in false colour in panels C and D of figure 5.

Additional Figure 5:

Figure 5. HR TEM images of nanocrystalline SiNP. Panels A and B show the agglomerated nature of the SiNPs and that the SiNPs are crystalline. Lattice fringes and d spacing's are indicated in yellow along with the orientations. Panels C and D shows the highlight the shape of various SiNPs in false color for panels A and B, respectively

Worse yet, the FTIR, Raman, and EDS all show Oxygen in the final products. The product is NOT pure Si.

We understand the reviewers point here. However, the data and the additional data as well as similar studies indicate a different interpretation. We have now modifications to clarify this issue.

For ease, the additional text now reads:

The obtained final product matches the quality of Si nanoparticles prepared in similar synthetic routes by Veinot et al. [6, 27] and Jaumann et al. [10]. Veinot et al. [27] obtained free-standing hydrogen-terminated silicon nanocrystals, as evidenced by the characteristic stretching and scissoring frequencies at ca. 2100 and 910 cm^{-1} , respectively from IR spectroscopic investigations. While, Jaumann et al. [10] obtained crystalline silicon nanoparticles with a size of 2-5 nm. They also noticed the tendency for the nanoparticles to agglomerate. Both groups also showed a small presence of oxygen in the obtained final product, which was attributed to residues and limited surface oxidation after sample preparation [6, 10, 27].

The presence of oxygen in the final product can be due in part to minute surface oxide formation and trapped oxygen species. This matches observations by others [6, 10, 27].

- *Even the high resolution TEM (Fig.4c) show one Si NP locally, and the XRD show Si Peaks, the quality of the products is really suspicious.*

We have not been clear previously as the reviewer highlights and we have now made changes as discussed above which are also relevant to this point and hope that it is now clearer. Firstly as pointed out now in the text and above, we show that our data (microscopy and XRD) matches exceedingly well with other fine SiNP data as published in the literature. For example references 6, 10 and 27 in the manuscript. Indeed we suggest we now, thanks to the reviewers and editors suggestions, have superior HR TEM microscopy studies showing greater details of the SiNPs. In particular we refer the reviewer to the newly added figure 5 which confirms the crystalline nature of the SiNPs and their morphology and clustering which is provided above in a previous response to the reviewer, along with additional text discussing the new TEM data (as well as SEM and optical microscopy data).

We feel that the manuscript is improved thanks to the reviewers constructive comments and hope it is now acceptable for publication.